# High-Fat-Diet-Induced Oxidative Stress in Giant Freshwater Prawn (*Macrobrachium rosenbergii*) via NF-κB/NO Signal Pathway and the Amelioration of Vitamin E

**DOI:** 10.3390/antiox11020228

**Published:** 2022-01-25

**Authors:** Cunxin Sun, Fan Shan, Mingyang Liu, Bo Liu, Qunlan Zhou, Xiaochuan Zheng, Xiaodi Xu

**Affiliations:** 1Key Laboratory of Freshwater Fisheries and Germplasm Resources Utilization, Ministry of Agriculture and Rural Affairs, Freshwater Fisheries Research Center, Chinese Academy of Fishery Sciences, Wuxi 214081, China; suncx@ffrc.cn (C.S.); zhouql@ffrc.cn (Q.Z.); zhengxiaochuan@ffrc.cn (X.Z.); 2Wuxi Fisheries College, Nanjing Agricultural University, Wuxi 214081, China; 15261865196@163.com (F.S.); 2018813045@njau.edu.cn (M.L.); 2021213005@stu.njau.edu.cn (X.X.)

**Keywords:** high fat diet, oxidative stress, NF-κB, vitamin E, *Macrobrachium rosenbergii*

## Abstract

Lipids work as essential energy sources for organisms. However, prawns fed on high-fat diets suffer from oxidative stress, whose potential mechanisms are poorly understood. The present study aimed to explore the regulation mechanism of oxidative stress induced by high fat and the amelioration by vitamin E (VE) of oxidative stress. *Macrobrachium rosenbergii* were fed with two dietary fat levels (LF 9% and HF 13%) and two VE levels (200 mg/kg and 600 mg/kg) for 8 weeks. The results showed that the HF diet decreased the growth performance, survival rate and antioxidant capacity of *M. rosenbergii*, as well as inducing hypertrophied lipid droplets, lipophagy and apoptosis. A total of 600 mg/kg of VE in the HF diet alleviated the negative effects induced by HF. In addition, the HF diet suppressed the expression of toll-dorsal and imd-relish signal pathways. After the relish and dorsal pathways were knocked down, the downstream iNOS and NO levels decreased and the MDA level increased. The results indicated that *M. rosenbergii* fed with a high-fat diet could cause oxidative damage. Its molecular mechanism may be attributed to the fact that high fat suppresses the NF-κB/NO signaling pathway mediating pro-oxidant and antioxidant targets for regulation of oxidative stress. Dietary VE in an HF diet alleviated hepatopancreas oxidative stress and apoptosis.

## 1. Introduction

For a long time, China’s aquaculture industry has been steadily pursuing growth and using high-protein feeds in intensive development, which has led to a series of problems, such as a decline in flesh quality, immunosuppression, and water eutrophication. As a more environmentally friendly nutrient than protein, fat has been shown to have significant protein-sharing effects [1,2]. Therefore, developing high-fat diets in the field of aquafeeds has broad application prospects. However, in current production practices, long-term feeding with high fat diets has led to problems such as oxidative stress and immunosuppression in cultured aquatic animals [3,4]. Vitamin E (VE) is a fat-soluble vitamin that can maintain the stability of the phospholipid bilayer of the cell membrane through antioxidant effects. When free radicals attack the biofilm, VE can undergo a redox reaction with the oxygen-containing groups of ROS to block the generation of peroxides. Studies have shown that VE can significantly enhance specific immunity and anti-stress effects in *Macrobrachium nipponense* [5] and *Palaemonetes argentinus* [6]. In addition, VE has a particular alleviating effect on oxidative stress caused by oxidized fat and high-fat diets [3,7]. Given this, it is crucial to study the mechanism of oxidative stress in organisms induced by high-fat diets and evaluate the amelioration of VE.

For farmed fish, oxidative stress damage induced by the long-term feeding of high-fat diets is manifested by a surge in inflammatory factors, decreased activity of antioxidant enzymes, abnormal mitochondrial metabolism, and increased apoptosis rate, which seriously affect growth performance, stress tolerance, and pathogen sensitivity [3,8]. Further research found that oxidative stress is mainly induced by the long-term free fatty acid level, which exceeds the metabolic capacity of the body, and leads to an increase in ROS production in the adipose tissue. On the other hand, adipose tissue oxidative stress induces abnormal cytokine production in fat. These two pathways form a vicious cycle in the body that aggravates lipid metabolism disorders and oxidative stress damage [9,10,11]. Fish and shrimp differ greatly in lipid digestion and metabolism. The major difference between shrimp and fish is that crustaceans do not produce bile and cannot use bile salts in fat digestion and metabolism [12]. Besides, fish accumulate fat in the hepatocyte, the adipose tissues in the abdominal cavity, and between the skin and flesh, while shrimp deposit fat mainly in the hepatocyte, which indicates an shrimps’ inability to tolerate and utilize higher dietary fat levels [13]. In the crustacean species, the haemolymph is an essential component of the immunological barrier system and nutrient exchange, which also performs wide functions, such as cellular and biochemical transport, oxygen exchange, and osmotic pressure regulation. The hepatopancreas is the primary organ responsible for the synthesis and degradation of fat, and performs digestion, enzyme secretion, and the excretion of waste materials, which is analogous to the liver in vertebrates. However, studies on haemolymph and hepatopancreas response, as well as oxidative stress induced by high-fat diets in crustaceans, have rarely been reported, with most research focusing on fish species. According to these limited studies, only malondialdehyde and antioxidant enzymes were measured in the hepatopancreas of crustaceans fed high-fat diets [14,15]. The regulatory mechanism of oxidative stress induced by high fat is yet to be determined.

As a critical transcription factor regulating inflammatory response, NF-κB has been shown to be closely related to immune regulation, cell cycle regulation, tumor metastasis, and apoptosis in vertebrates [16,17]. Dorsal and relish are two homologous proteins of NF-κB found in mollusks and crustaceans [18,19]. The main upstream and downstream factors of NF-κB in shrimp have also been determined. The stimulatory and inhibitory roles of NF-κB in ROS regulation have been investigated in vertebrates [20]. However, the studies of the NF-κB signaling pathway in shrimp mainly focus on the immune response to bacteria or viruses [21,22,23,24]. Few studies focus on oxidative stress regulation, which is yet to be further explored and verified.

The giant freshwater prawn *Macrobrachium rosenbergii* has been widely cultivated in China and worldwide. Its output in China was about 161,888 tons in 2020, with an increase of 15.96% over the previous year [25]. In the process of breeding, *M. rosenbergii* generally has high feed protein (> 40%) and high density of breeding, which causes significant feed waste and water eutrophication [26]. Therefore, this species can be used as an ideal model for developing high-fat diets and exploring the mechanism of oxidative stress in invertebrates. In view of this, the present study aimed to evaluate the molecular mechanism of oxidative stress induced by high-fat diets in *M. rosenbergii* and the amelioration of VE. The results will shed more light on the molecular mechanism of oxidative stress induced by nutrients, and also provide a reference for the development and application of high-fat feed.

## 2. Materials and Methods

### 2.1. Experimental Ingredients and Diets

Four experimental diets were formulated in this study, including two levels of fat and VE. LF/200 VE included 9% fat and 200 mg/kg VE, LF/600 VE included 9% fat and 600 mg/kg VE, HF/200 VE included 13% fat and 200 mg/kg VE, an HF/600 VE included 13% fat and 600 mg/kg VE. The formulation and proximate composition of the experimental diets are presented in Table 1. Fish meal, casein, and gelatin served as the protein sources; fish oil was supplemented as the lipid source; and α-starch and dextrin were used as the carbohydrate source.

All the ingredients were ground through a 60 mm mesh. The fine powder was carefully weighed, then lipid sources and 30% water were added to the mixture, which was further blended to ensure homogeneity. A laboratory pelletizer (Guangzhou Huagong Optical Mechanical and Electrical Technology CO. LTD, Guangzhou, China) was used for the pelletizing process. The diet diameter was 1.5 mm. After drying in the laundry drier, the feeds were offered to prawns.

### 2.2. Prawns and the Feeding Trial

Juvenile prawns were provided by Zhejiang Southern Taihu Lake freshwater aquatic seed industry CO. LTD (Huzhou, China). After two weeks of acclimation, prawns of similar size (0.24 ± 0.001 g) were randomly distributed into 12 concrete tanks (2.0 m × 1.5 m × 0.8 m) at a rate of 50 prawns per tank. Four experimental diets were randomly allotted to prawns with triplicate tanks. All prawns were fed three times daily at 7:00, 12:00, and 17:30 for 56 days, and the feeding rate was 2–5% of body weight. Feces and molts were removed by siphoning the tanks. During the feeding trial, the average water temperature was 30 ± 0.4 °C; continuous aeration was supplied to each tank to maintain the dissolved oxygen above 50 mg/L; pH was 7.6–8.0, and total ammonia nitrogen level was above 0.02 mg/L. Total ammonia nitrogen and nitrite were kept < 0.2 and 0.005 mg/L, respectively.

### 2.3. Sample Collection

At the end of the feeding period, prawns were fasted for 24 h to empty the digestive tract. The hemolymphs from five prawns were randomly sampled from the cardiocoelom per tank. Alsever’s solution was used as the anticoagulant at a ratio of 1:1 with haemolymph. Hemolymph samples were collected into anticoagulation tubes, then centrifuged at 2000× *g* 4 °C for 10 min. Hepatopancreas was stored in 4% paraformaldehyde for apoptosis measurement and 2.5% glutaraldehyde for ultrastructure study. Furthermore, the remaining hepatopancreases were quickly removed and stored at −80 °C for subsequent analysis.

### 2.4. Ultrastructure Study

Electron microscopy samples were fixed with 2.5% glutaraldehyde for 24 h, then fixed with 1% osmium tetroxide for 1 h and stored at 4 °C. The sections were embedded in epoxy resin Epon 812, cut into 70 nm-thick sections by RMC PowerTomeXL microtome, stained with uranyl acetate and lead citrate, and examined under a transmission electron microscope (Hitachi H-7650, Tokyo, Japan).

### 2.5. Apoptosis Detection

Hepatocyte apoptosis was determined by Lu et al.’s methods [27], the terminal deoxynucleotidyl transferase-mediated dUTP-biotin nick end labeling (TUNEL) assay followed the protocol of the apoptosis detection kit (Nanjing Jian-Cheng Bioengineering Institute, Nanjing, China). The positive cell nucleus was dyed brown-yellow granules. The DNase1-treated tissue was used as the positive control. The reaction without TdT enzyme was used as the negative control.

### 2.6. Biochemical and Antioxidative Parameters in Hemolymph

Total cholesterol (TC), total triglycerides (TG), alanine aminotransferase (ALT) and aspartate aminotransferase (AST) were determined by an automatic hemolymph biochemical analyzer (Mindray BS-400, Shenzhen, China), as described by Wangari et al. [28]. Inducible nitric oxide synthase (iNOS), superoxide dismutase (SOD), glutathione peroxidase (GPx) and anti-superoxide anion (ASA) activity, as well as malonaldehyde (MDA) nitric oxide (NO) content in hemolymph were detected using a commercially available assay kit (Nanjing Jiancheng Bioengineering Institute, Nanjing, China) according to the manufacturer’s instructions.

### 2.7. RNA Isolation and RT-qPCR Analysis

Total RNA was isolated using RNAiso Plus (Takara Co. Ltd., Tokyo, Japan), and then purified with RNase-Free DNase (Takara Co. Ltd., Tokyo, Japan) to avoid genomic DNA amplification. The purity and concentration of RNA were measured using a NanoDrop (DN-1000, Thermo Scientific, Waltham, MA, USA). After normalizing the concentration of the RNA samples, cDNA was generated from 500 ng DNase-treated RNA using ExScriptTM RT-PCR kit, according to the manufacturer’s instructions (Takara Co. Ltd., Tokyo, Japan).

The cDNA samples were analyzed by real-time quantitative detector (BIO-RAD, Hercules, CA, USA) using a SYBR Green II Fluorescence Kit (Takara Co. Ltd., Tokyo, Japan). The fluorescent qPCR reaction solution consisted of 10 μL SYBR^®^ premix Ex TaqTM, 0.4 μL ROX Reference Dye II, 0.4 μL PCR forward primer (10 μM), 0.4 μL PCR reverse primer (10 μM), 2.0 μL RT reaction (cDNA solution), and 6.8 μL dH2O. All RT-qPCR primers were designed using Primer 5 software and listed in Table 2. The thermal profile was 95 °C for 30 s, followed by 40 cycles of 95 °C for 5s and 60 °C for 30 s, followed by a melt curve analysis of 15 s from 95 to 60 °C, 1 min for 60 °C, and then up to 95 °C for 15 s. Control reactions were conducted with non-reverse transcribed RNA to determine the level of background or genomic DNA contamination, respectively. In all cases, genomic DNA contamination was negligible. β-actin was selected as the housekeeping gene to normalize our samples because of its stable expression in the present study. The reaction was carried out in three duplicates of each sample. Values for the threshold (CT) from the treated and control tissue templates were compared, and the 2^−^^ΔΔCT^ method was used as the relative quantification calculation method.

### 2.8. Knock-Down of Relish and Dorsal In Vivo Expression by RNA Interference

To determine the duration and efficiency of gene knock-down of silencing, *M. rosenbergii* were divided into four groups. The first group was a blank group and PBS was injected; the second group was the negative control group and 1.5 μg/g dsGFP was injected; the third group was the dorsal group and 1.5 μg/g dsdorsal was injected; the fourth group was the relish group and 1.5 μg/g dsrelish was injected. Hepatopancreases were harvested at 1 d, 4 d, 7 d, and 10 d after injection. The efficiency of gene knock-down was monitored using RT-qPCR analysis with primers T7-*relish* and T7-*dorsal*. β-actin expression analysis was used as an internal control. The optimum time point of gene silencing was found to be at 4 d after dsRNA injection, and the duration is 7 days after injection. In all of the following experiments, the prawns were therefore injected with dsRNA consecutive 4 day intervals. This knockdown assay was carried out with three replicates.

In total, 12 prawns from the LF/200VE and 36 prawns from the HF/200VE were divided into 4 groups for intramuscular injection. LF/200VE group was injected with 1.5 μg/g dsGFP; HF/200VE group was injected with 1.5 μg/g dsGFP; dsdorsal and dsrelish, respectively. After 24 h of injection, the first group was fed the basal diet, and the other groups were fed with the high-fat diet for 2 weeks. After feeding trial, the hemolymph and hepatopancreases were sampled for analysis.

### 2.9. Calculations and Statistical Analysis

The growth parameter was calculated as follows:Weight gain rate (WGR, %) = (W_t_ − W_0_) × 100/W0.Specific growth rate (SGR, % day^−1^) = (LnW_t_ − LnW_0_) × 100/day.Feed conversion ratio (FCR) = feed consumption (g)/Weight gain (g).Survival rate (SR, %) = initial number/final number × 100Where W_0_ and W_t_ are initial and final body weight.

Data were subjected to two-way analysis of variance (ANOVA) to investigate the growth performance, hemolymph parameters and mRNA expression, after testing the homogeneity of variances with the Levene test. If significant (*p* < 0.05) differences were found, Duncan multiple range test was used to rank the means. Analyses were performed using the SPSS program v16.0 (SPSS Inc., Michigan Avenue, Chicago, IL, USA) for Windows. All data were presented as means ± S.E.M (standard error of the mean).

## 3. Results

### 3.1. Growth Performance

As shown in Figure 1, dietary fat and VE level had significant effects on the growth performance and survival rate of *Macrobrachium rosenbergii*, and the interaction of dietary fat and VE levels was significant (*p* < 0.05). In the prawns fed LF diets, different dietary levels of VE had no significant effect on SR, WGR, SGR, and FCR (*p* > 0.05). In the prawns fed with 200VE diets, SR, WGR, and SGR decreased significantly with increasing dietary fat level (*p* < 0.05), and FCR increased significantly (*p* < 0.05). In the prawns fed with HF diets, SR, WGR, and SGR increased significantly with increasing dietary VE levels (*p* < 0.05), and FCR reduced significantly (*p* < 0.05).

### 3.2. Hemolymph Biochemistry Parameters

As shown in Figure 2, hemolymph AST and TC were significantly affected by dietary fat levels (*p* < 0.05), TC was significantly affected by dietary VE levels, and AST and TG were significantly affected by the interaction of dietary fat and VE levels (*p* < 0.05). In prawns fed LF diets, the levels of dietary VE showed no significant effect on the levels of ALT, AST, TC, and TG (*p* < 0.05). In the prawns fed 200VE diets, ALT, AST, TC, and TG levels increased significantly with increasing dietary fat levels (*p* < 0.05). In the prawns fed HF diets, ALT, AST, TC, and TG levels decreased significantly with increasing dietary VE levels (*p* < 0.05).

### 3.3. Hepatopancreas Ultrastructure and Apoptosis

The prawn hepatopancreas ultrastructure pictures are presented in Figure 3. The prawns fed the LF diets exhibited normal ultrastructures with round and clear nuclei; fewer lipid droplets were visible. The 600 VE group exhibited increased endoplasmic reticulum compared with the 200 VE group. Nevertheless, the prawns fed the HF diet showed hypertrophied lipid droplets in the cytoplasm, and lipophagy was observed. A total of 600 VE in the HF diet led to alleviated hepatic lipid accumulation with fewer droplets surrounding the centered nucleus compared with the 200 VE group.

The apoptosis of the hepatopancreatic cells is shown in Figure 4. The cells with stained brown nuclei were considered in a state of apoptosis and were counted to calculate the hepatopancreatic apoptosis cell ratio. Three paraffin sections were used for the TUNEL assay in each group. In the prawns fed LF diets, apoptotic cells were about 16–17% of the total hepatocytes (Figure 4A,B). In the hepatopancreases from the prawns fed the HF/200 VE diet, the apoptotic cells were about 30% of the total hepatocytes, which was significantly higher than in those fed LF/200 VE diet (*p* < 0.05) (Figure 4C). The 600 VE addition significantly decreased the ratio of apoptosis cells compared with 200 VE in the HF groups (*p* < 0.05) (Figure 4D).

### 3.4. Hemolymph Antioxidant Capacity

The hemolymph antioxidant indexes are shown in Figure 5. SOD, GPX, and ASA were significantly affected by dietary fat levels (*p* < 0.05), GPx was significantly affected by dietary VE levels, and SOD was significantly affected by the interaction of dietary fat and VE (*p* < 0.05). In prawns fed LF diets, dietary levels of VE had no significant effect on the levels of SOD, GPx, ASA, and MDA (*p* < 0.05). In the prawns fed 200VE diets, the SOD, GPx, and ASA activities were significantly reduced when the dietary fat level was increased, while the MDA content was significantly increased (*p* < 0.05). In the prawns fed HF diets, the GPx activity was significantly increased and the MDA level was significantly decreased when the VE level was increased (*p* < 0.05).

The hemolymph iNOS activity and NO content are shown in Figure 6. The iNOS and NO were significantly affected by dietary fat levels (*p* < 0.05) and iNOS was significantly affected by the interaction of dietary fat and VE (*p* < 0.05). In the prawns fed 200 VE diets, the iNOS and NO levels were significantly reduced when the dietary fat level was increased (*p* < 0.05). In prawns fed the same fat-level diets, VE levels showed no significant difference in iNOS activity or NO content (*p* < 0.05).

### 3.5. Hepatopancreas NF-κB Signal Pathway Expression

The hemolymph gene expression of the NF-κB signal pathway expression is shown in Figure 7. *Imd*, *relish*, *toll*, and *dorsal* expression were significantly affected by dietary fat levels (*p* < 0.05), *Imd*, *relish, dorsal* expression was significantly affected by dietary VE levels (*p* < 0.05), and the interaction of dietary fat and VE showed no significant difference in NF-κB signal pathway expression (*p* > 0.05). In the prawns fed LF diets, dietary 600 VE significantly inhibited the expression of *imd* and *dorsal* compared with 200 VE (*p* < 0.05). In the prawns fed 200 VE diets, dietary HF significantly inhibited the expression of *imd, relish, toll*, and *dorsal* compared with LF (*p* < 0.05). In the prawns fed HF diets, dietary 600 VE significantly inhibited *imd* and *relish* expression compared with 200VE (*p* < 0.05). No significant difference was observed in the expression of *toll* or *dorsal* (*p* > 0.05).

### 3.6. In Vivo Knock-Down of Relish and Dorsal by RNA Interference

A relish and dorsal knock-down experiment was performed to further characterize the role of NF-κB in the antioxidative process induced by dietary HF. As shown in Figure 8A,B, *relish* and *dorsal* expression reduced significantly at 4d–7d post-injection of *relish*- and *dorsal*-specific dsRNA (*p* < 0.05). After a two-week in vivo knock-down of *relish* and *dorsal* by RNA interference, the expression of *dorsal* was suppressed by the *dorsal* dsRNA injection (*p* < 0.05) (Figure 8C), and the expression of *relish* was suppressed by the *relish* dsRNA injection (*p* < 0.05) (Figure 8D).

### 3.7. Hemolymph Antioxidant Capacity after NF-κB Suppression

As shown in Figure 9, the iNOS activity and NO content in the hemolymph of *M. rosenbergii* fed with the HF diet significantly decreased (*p* < 0.05). A further decline in the iNOS and NO levels was observed when the expression of *relish* and *dorsal* was suppressed (*p* < 0.05). Hemolymph ASA activity and MDA content are showed in Figure 10. Compared with the control group, the HF group’s MDA content significantly increased and its ASA activity significantly decreased (*p* < 0.05). In the prawns fed HF diet, *relish* and *dorsal* suppression further increased the MDA content and decreased the ASA activity (*p* < 0.05).

## 4. Discussion

Physiological conditions limit carnivorous fish’s demand for and utilization of fat. The long-term intake of high amounts of fat will accumulate in the body, leading to obstacles in fat transport, affecting body tissue fat, and reducing body fat deposition and fat metabolism by regulating lipid metabolism. Improving fish health has become one of the feasible means. In this study, the high-fat diet reduced the growth performance and survival rate of *M. rosenbergii*. Similar results were also observed in spotted seabass (*Lateolabrax maculatus*) [29], Pacific white shrimp (*Litopenaeus vannamei*) [14], largemouth bass (*Micropterus salmoides*) [30], and blunt snout bream (*Megalobrama amblycephala*) [31]. The reason for this might be the damaged antioxidant capacity, stress and disease resistance induced by dietary high fat. The previous studies pointed out that a high-fat diet could induce aberrant hepatic lipid secretion, reduce mortality rates and worsen the adverse effects of antibiotics by activating oxidant stress and endoplasmic reticulum stress [32,33,34]. This is also supported by the fact that high fat reduced the survival rate of *M. rosenbergii* in the present study. VE supplementation alleviated the growth inhibition induced by high fat levels in the present study, which was also confirmed in turbot (*Scophthalmus maximus*) [3]. Previous studies found that VE could alleviate high-fat-diet-induced hepatic oxidative stress and hypoimmunity to keep health and growth, while VE deficiency inhibited fat metabolism and induced lipid peroxidation [3,35]. The above results suggested that VE may be beneficial for maintaining the growth and health of prawns by reducing lipid metabolism disorder and the oxidative stress induced by high-fat diets.

Haemolymph biochemical indicators are important in diagnosing the health of aquatic animals. ALT and AST are sensitive biomarkers for hepatopancreas injury, and the TC and TG content can reflect lipid metabolism. In this study, the increase in the hemolymph ALT and AST activities of *M. rosenbergii* fed with a high-fat diet indicated the impaired function and metabolism of the hepatopancreas. Similar results were also observed in blunt snout bream [27] and grass carp (*Ctenopharyngodon idella*) [36]. The increase in hemolymph TC and TG in high fat group also indicated lipid metabolic disturbances and disorders because high serum TG and TC are significant risk factors related to fatty liver that can result in oxidative stress and decrease disease resistance. The present study found that extra dietary VE reduced the hemolymph levels of TG and TC, as well as ALT and AST activities in prawns fed an HF diet. The previous study demonstrated that the appropriate amounts of dietary VE improve the metabolic health of the liver in grass carp juveniles [37]. This was further supported by the fact that the VE requirement of aquatic animals increases as the fat content of feed increases [38]. These findings suggested that extra VE supplementation may contribute to the hepatopancreas health benefits of *M. rosenbergii*.

The hepatopancreas is the main organ responsible for the absorption and storage of the ingested substances. In the present study, the HF diet caused excessive lipid droplet accumulation in the hepatopancreas, which was also widely reported in other aquatic species [27,29,39]. In addition, the high-fat diet also induced hepatopancreas lipophagy and apoptosis. Lipophagy is the selective autophagy of lipid droplets degraded by lysosomes, which is associated with the regulation of lipid metabolism [40]. Existing evidence shows that high-fat diets can induce endoplasmic reticulum stress, autophagy, and apoptosis [41]. These pathologies may also cause the abnormal secretion of lipoproteins and lipid peroxidation, leading to a vicious circle. The results obtained here are still highly significant for human beings, although there are huge differences in the physiological structure of aquatic and terrestrial animals. In mice, high-fat diets could induce dyslipidemia and non-alcoholic fatty liver disease (NAFLD) [42,43]. Nutritional liver disease in shrimp has not yet been clearly defined, but the apoptosis and steatosis of the hepatopancreas found in prawns fed high levels of fat in the present study were similar to the NAFLD symptoms in humans. Nevertheless, these deformities can be relieved by extra VE supplementation. According to previous studies, VE can protect cells against lipid peroxidation and attenuate hepatic steatosis and mitochondrial damage [44]. VE could ameliorate apoptosis and autophagy induced by stress and disease in mammals [45,46,47]. Thus, based on histological and TUNEL apoptosis observation, the lipid-lowering effect of VE may be attributed to the attenuation of apoptosis and autophagy induced by high-fat diets. Similar results were also found in rats [48], indicating that *M. rosenbergii* can serve as a model for developing drugs for NAFLD.

Oxidative stress damage refers to the excess reactive oxygen species (ROS) production stimulated by factors such as the environment and nutrition, which breaks the redox equilibrium and ultimately leads to cell and tissue damage. In the present study, the HF diet reduced hemolymph ASA activity and increased MDA content. Similar results were also observed in largemouth bass (*Micropterus salmoides*) [49] and *M. amblycephala* [8]. This indicates that excess fat might induce oxidative stress damage and lipid peroxidation as MDA is the main component of lipid peroxide, which damages cell structures and functions [3]. In addition, the antioxidant enzyme system composed of SOD and GPx can remove excessive free radicals, reducing the damage by lipid peroxidation. The inhibition of the SOD and GPx activities in the HF group further supported the notion that excess fat induced oxidative stress. Nevertheless, VE improved GPx activities and reduced MDA content in the prawns fed the high-fat diet. This might due to the fact that VE has a significant effect of enhancing specific immunity and anti-oxidants by undergoing a redox reaction with the oxygen-containing groups of ROS to block peroxide production [50]. The ameliorative effect of VE on oxidative stress and hypoimmunity induced by a high-fat diet and oxidized oil also supported these results [3,51]. NO generated by iNOS could inhibit lipid peroxidation, which relies on its ability to eliminate oxygen free radicals [52]. The inhibition of hemolymph iNOS and NO in prawns fed with high fat also indicated the oxidative damage induced by excess fat, which is supported by the results in mussels [53] and *L. vannamei* [54]. However, VE showed no significant effect on iNOS and NO. As a non-enzymatic oxidant, VE halts lipid peroxidation by donating its phenolic hydrogen to peroxyl radicals, forming tocopherol radicals [55]. This may partly explain the reason of the invalidation of VE on iNOS and NO levels.

In general, the activation of transcription factors NF-κB is an indispensable step for increasing iNOS activity, which subsequently induces the release of NO [56]. In the present study, the expression of *imd*-*relish* and *toll*-*doral*, two major homologs of the components of NF-κB signal pathways, were inhibited by dietary high fat. It is speculated that both the *imd*-*relish* and the *toll*-*doral* signal pathway were suppressed by oxidative stress, which further reduced the production of NO, forming a vicious circle. A previous study also supported the notion that oxidative stress induced by high fat could inhibit the expression of the toll signal pathway [3]. However, VE only decreased *imd*-*relish* expression in the HF group. The regulation of VE in high-fat diets might attribute to the production of anti-microbial peptides as *relish* translocates into the nucleus to activate the expression of antibacterial peptide genes after stimulation [57]. To further verify the effect of the regulation of NF-κB on oxidative stress induced by high levels of fat, *relish* and *dorsal* expression were knocked down, respectively. The levels of iNOS and NO were suppressed, while MDA content increased and ASA activity decreased after relish and dorsal knock-down. The results above suggested that dietary high fat induced oxidative stress via the suppression of the NF-κB/NO signal pathway. Some research supported the notion that the NF-κB signal pathway also plays a crucial regulatory role in NAFLD induced by high-fat diets [58,59]. In vertebrates, NF-κB is activated by high-fat diets to induce inflammation. Nevertheless, crustaceans lack inflammatory cytokines. The different regulation of NF-κB in crustaceans and vertebrates still requires further in-depth studies.

## 5. Conclusions

The present study investigated the growth performance and oxidative status of *M. rosenbergii* fed different fat and VE levels, and revealed the molecular regulatory mechanism in response to a high-fat diet. In the present study, VE ameliorated the growth retardation and oxidative stress induced by a high-fat diet. A putative mechanism that could explain our results is shown in Figure 11. The high-fat diet caused excessive lipid deposition in *M. rosenbergii*, which further induced hepatopancreas lipophagy, apoptosis, and lipid peroxidation, resulting in oxidative stress damage. The *toll-dorsal* and *imd-relish* signaling pathways were inhibited by the high fat levels, which mediated pro-oxidant targets (iNOS and NO) and antioxidant targets (SOD and GPx) for the negative feedback regulation of oxidative stress. Dietary VE supplementation combined with the high-fat diet alleviated hepatopancreas oxidative stress and apoptosis. The reason for this may be related to the antioxidant properties of VE and the activation of the antioxidant enzyme system.

## Figures and Tables

**Figure 1 antioxidants-11-00228-f001:**
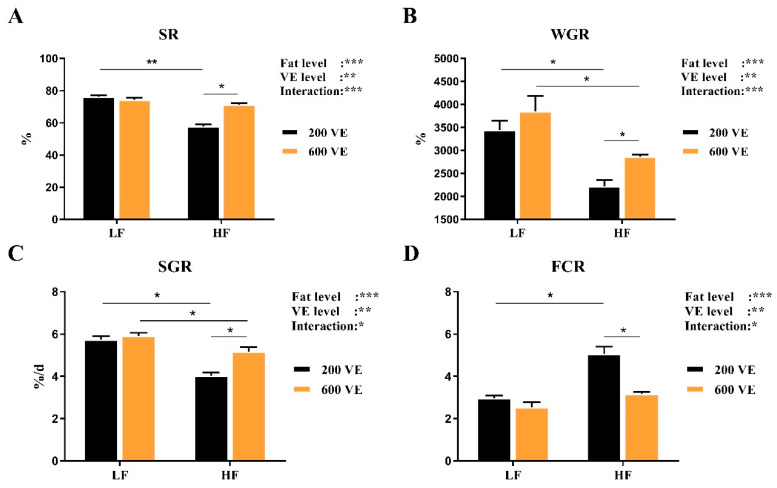
Growth performance and feed utilization of *Macrobrachium rosenbergii* fed with different dietary fat and VE levels. (**A**) survival rate (SR, %) of *M. rosenbergii* fed with different dietary fat and VE levels; (**B**) Weight gain rate (WGR, %) of *M. rosenbergii* fed with different dietary fat and VE levels; (**C**) specific growth rate (SGR, %/day) of *M. rosenbergii* fed with different dietary fat and VE levels; (**D**) feed conversion ratio (FCR) of *M. rosenbergii* fed with different dietary fat and VE levels. Values are means ± SEM. ***: *p* < 0.001, **: *p* < 0.01, *: *p* < 0.05. WGR = (final weight − initial weight) × 100/initial weight; SGR = (Ln final weight − Ln initial weight) × 100/day; FCR = feed consumption (g)/weight gain (g); SR = final number of prawns/initial number of prawns.

**Figure 2 antioxidants-11-00228-f002:**
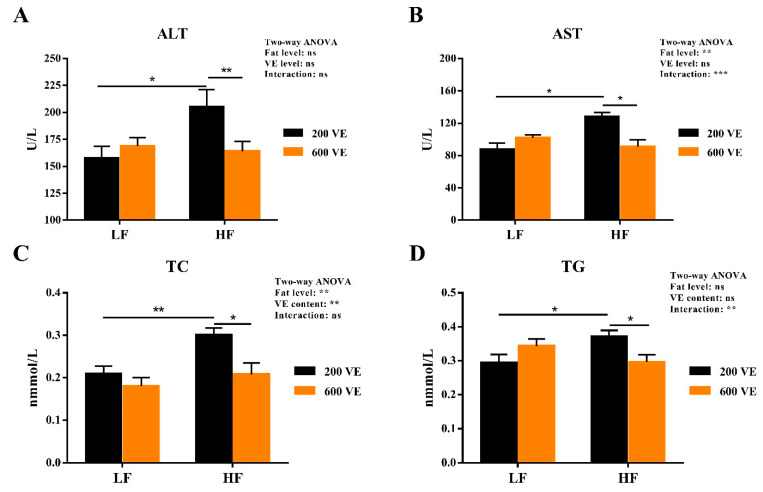
Hemolymph biochemistry parameters of *Macrobrachium rosenbergii* fed with different dietary fat and VE levels. (**A**) hemolymph alanine aminotransferase (ALT) activity of *M. rosenbergii* fed with different dietary fat and VE levels; (**B**) hemolymph aspartate aminotransferase (AST) activity of *M. rosenbergii* fed with different dietary fat and VE levels; (**C**) hemolymph total cholesterol (TC) content of *M. rosenbergii* fed with different dietary fat and VE levels; (**D**) hemolymph total triglycerides (TG) content of *M. rosenbergii* fed with different dietary fat and VE levels. Values are means ± SEM. ***: *p* < 0.001, **: *p* < 0.01, *: *p* < 0.05, ns: *p* < 0.05.

**Figure 3 antioxidants-11-00228-f003:**
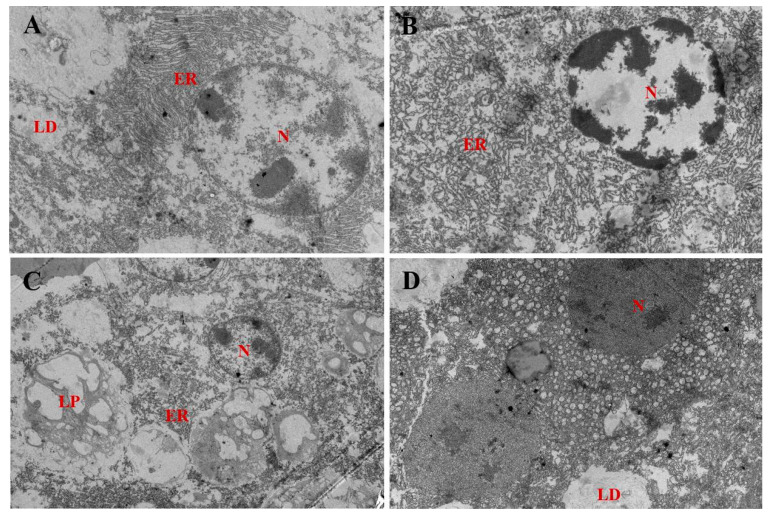
Transmission electron microscope images of *Macrobrachium rosenbergii* hepatocyte ultrastructure (2500×) fed with different dietary fat and VE levels. (**A**) prawns fed diet including 9% fat and 200 mg/kg VE; (**B**) prawns fed diet including 9% fat and 600 mg/kg VE; (**C**) prawns fed diet including 13% fat and 200 mg/kg VE; (**D**) prawns fed diet including 13% fat and 600 mg/kg VE. N: nucleus; LD: lipid droplet; ER, endoplasmic reticulum; LP, lipophagy.

**Figure 4 antioxidants-11-00228-f004:**
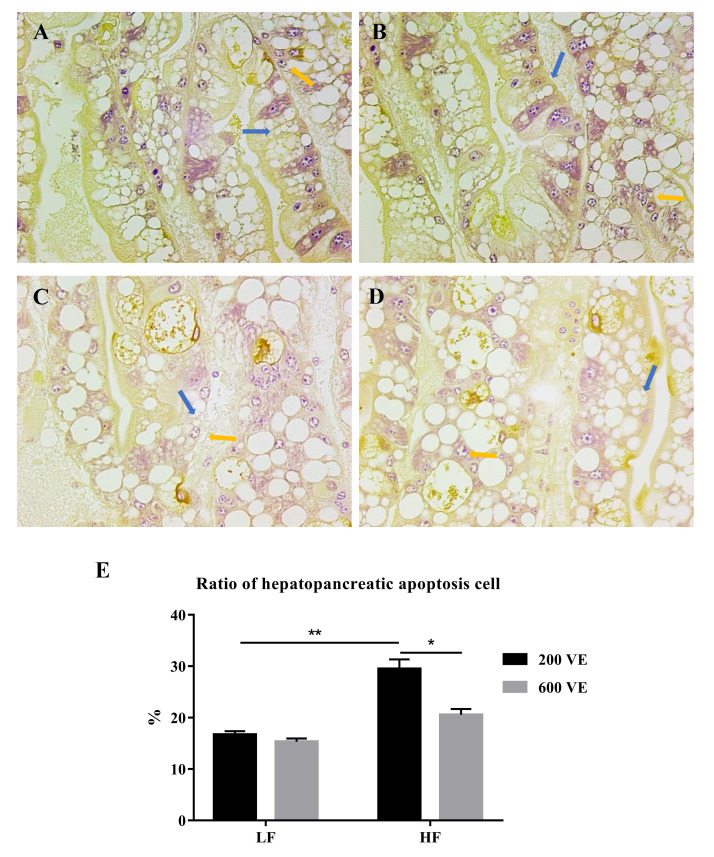
Hepatocyte apoptosis of *Macrobrachium rosenbergii* fed with different dietary fat and VE levels. (**A**) prawns fed diet including 9% fat and 200 mg/kg VE; (**B**) prawns fed diet including 9% fat and 600 mg/kg VE; (**C**) prawns fed diet including 13% fat and 200 mg/kg VE; (**D**) prawns fed diet including 13% fat and 600 mg/kg VE; (**E**) ratio of hepatopancreatic apoptosis cells (*n* = 3). Blue arrow: normal cells; yellow arrow: apoptotic cells. **: *p* < 0.01, *: *p* < 0.05.

**Figure 5 antioxidants-11-00228-f005:**
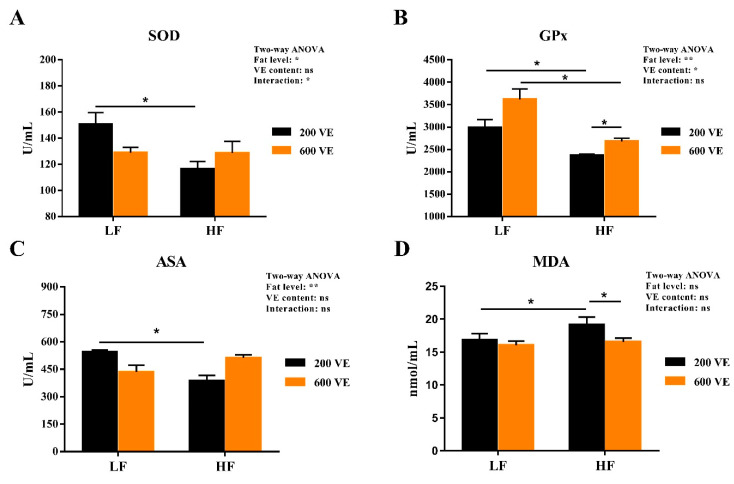
Hemolymph oxidative status of *Macrobrachium rosenbergii* fed with different dietary fat and VE levels. (**A**) hemolymph superoxide dismutase (SOD) activity of *M. rosenbergii* fed with different dietary fat and VE levels; (**B**) hemolymph glutathione peroxidase (GPx) activity of *M. rosenbergii* fed with different dietary fat and VE levels; (**C**) hemolymph anti-superoxide anion (ASA) activity of *M. rosenbergii* fed with different dietary fat and VE levels; (**D**) hemolymph malonaldehyde (MDA) content of *M. rosenbergii* fed with different dietary fat and VE levels. Values are means ± SEM. **: *p* < 0.01, *: *p* < 0.05, ns: *p* < 0.05.

**Figure 6 antioxidants-11-00228-f006:**
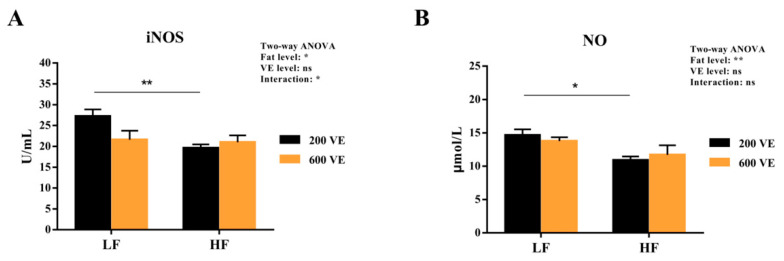
Hemolymph iNOS activity and NO content of *Macrobrachium rosenbergii* fed with different dietary fat and VE levels. (**A**) hemolymph inducible nitric oxide synthase (iNOS) activity of *M. rosenbergii* fed with different dietary fat and VE levels; (**B**) hemolymph nitric oxide (NO) content of *M. rosenbergii* fed with different dietary fat and VE levels. Values are means ± SEM. **: *p* < 0.01, *: *p* < 0.05, ns: *p* < 0.05.

**Figure 7 antioxidants-11-00228-f007:**
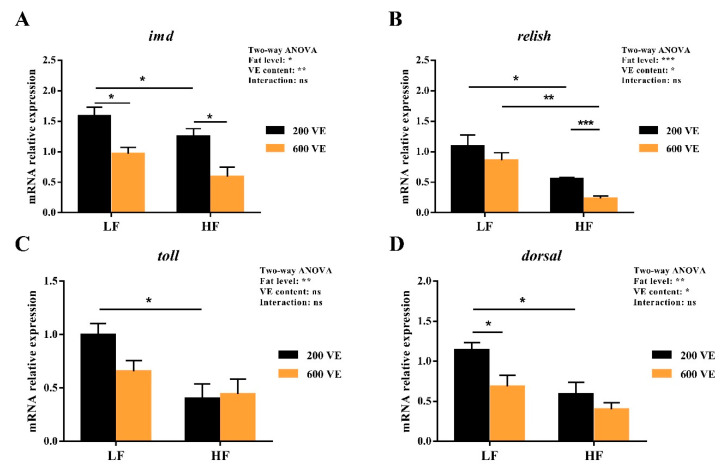
Relative expression of NF-κB signal pathway in hepatopancreas. (**A**) relative expression of *imd* in hepatopancreas; (**B**) relative expression of *relish* in hepatopancreas; (**C**) relative expression of *toll* in hepatopancreas; (**D**) relative expression of *dorsal* in hepatopancreas. Values are means ± SEM. ***: *p* < 0.001, **: *p* < 0.01, *: *p* < 0.05, ns: *p* < 0.05.

**Figure 8 antioxidants-11-00228-f008:**
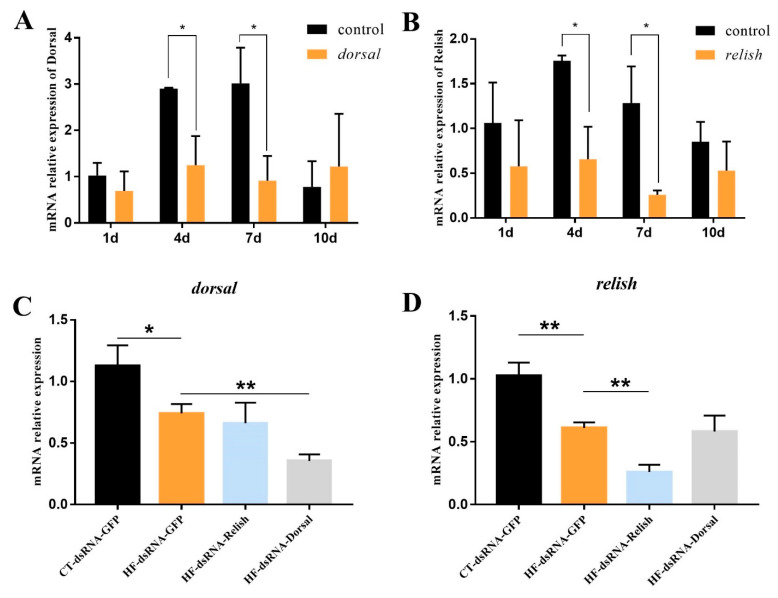
Knock-down of *relish* and *dorsal* in vivo expression by dsRNA-mediated RNA interference. (**A**) Time-course study of *dorsal* expression in hepatopancreas after RNA interference; (**B**) Time-course study of *relish* expression in hepatopancreas after RNA interference; (**C**) *dorsal* expression in *M. rosenbergii* fed with high-fat diet after RNA interference; (**D**) *relish* expression in *M. rosenbergii* fed with high-fat diet after RNA interference. Values are means ± SEM. **: *p* < 0.01, *: *p* < 0.05.

**Figure 9 antioxidants-11-00228-f009:**
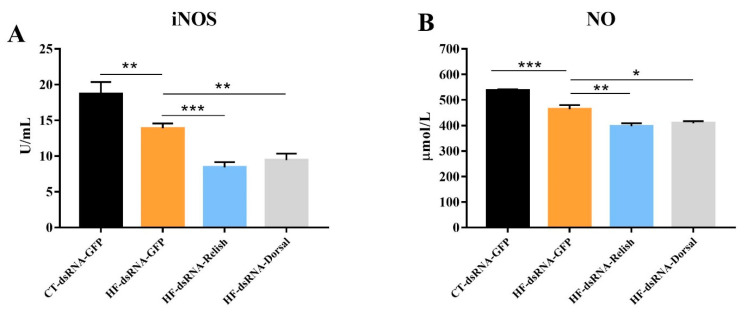
Hemolymph iNOS activity and NO content of *Macrobrachium rosenbergii* after knock-down by dsRNA-mediated RNA interference. (**A**) Hemolymph inducible nitric oxide synthase (iNOS) activity of *M. rosenbergii* after RNA interference; (**B**) Hemolymph nitric oxide (NO) content of *M. rosenbergii* after RNA interference. Values are means ± SEM. ***: *p* < 0.001, **: *p* < 0.01, *: *p* < 0.05.

**Figure 10 antioxidants-11-00228-f010:**
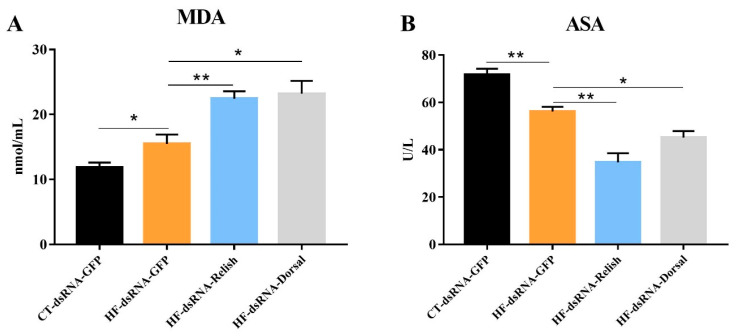
Hemolymph ASA activity and MDA content of *Macrobrachium rosenbergii* after knock-down by dsRNA-mediated RNA interference. (**A**) Hemolymph anti-superoxide anion (ASA) activity of *M. rosenbergii* after RNA interference; (**B**) Hemolymph malonaldehyde (MDA) content of *M. rosenbergii* after RNA interference. Values are means ± SEM. **: *p* < 0.01, *: *p* < 0.05.

**Figure 11 antioxidants-11-00228-f011:**
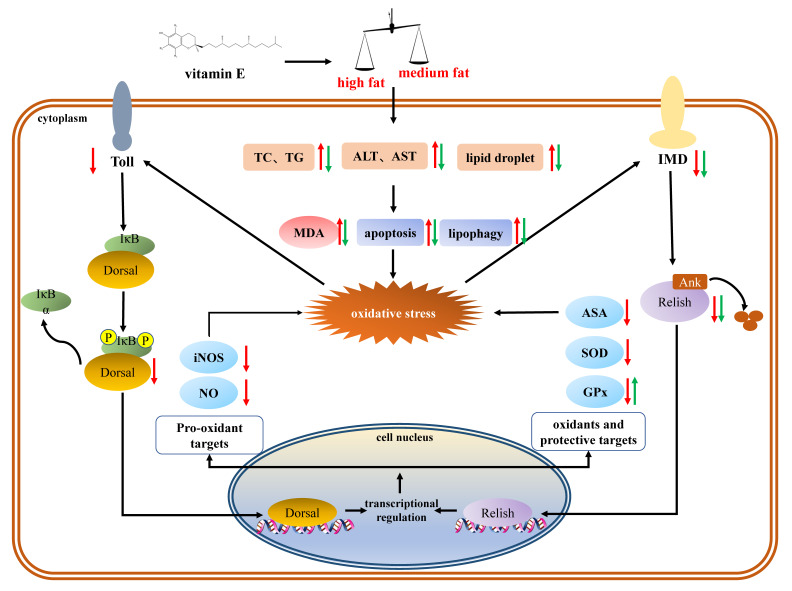
NF-κB/NO signaling mechanism in response to dietary high fat and VE. The red arrows indicate the effects of high fat; the green arrows indicate the effects of VE.

**Table 1 antioxidants-11-00228-t001:** Formulation and proximate composition of the experimental diets.

	LF/200VE	LF/600VE	HF/200VE	HF/600VE
Ingredients%				
Fish meal ^1^	22.00	22.00	22.00	22.00
Casein ^2^	24.00	24.00	24.00	24.00
Gelatin ^1^	6.00	6.00	6.00	6.00
α-starch ^3^	20.00	20.00	20.00	20.00
Dextrin ^3^	5.00	5.00	5.00	5.00
Fish oil ^1^	6.00	6.00	10.00	10.00
Soybean oil ^1^	0.00	0.00	0.00	0.00
Rapeseed oil ^1^	0.00	0.00	0.00	0.00
Microcystalline cellulose ^4^	5.00	5.00	1.00	1.00
Carboxymethyl cellulose ^4^	3.00	3.00	3.00	3.00
Bentonite ^1^	1.53	1.49	1.53	1.49
Soybean lecithin (50%) ^1^	2.00	2.00	2.00	2.00
Cholesterol ^1^	0.30	0.30	0.30	0.30
Ecdysone (2%) ^5^	0.10	0.10	0.10	0.10
DMPT ^5^	0.05	0.05	0.05	0.05
Choline chloride ^5^	1.00	1.00	1.00	1.00
Vitamin premix ^5^	1.00	1.00	1.00	1.00
Vitamin E ^1^	0.02	0.06	0.02	0.06
Mineral premix ^5^	1.00	1.00	1.00	1.00
Calcium dihydrogen phosphate ^1^	2.00	2.00	2.00	2.00
Proximate composition %				
Crude protein	40.90	40.90	40.90	40.90
Crude lipid	9.58	9.58	13.28	13.28
Gross energy (MJ/kg)	15.80	15.80	17.38	17.38

Notes: ^1^ Obtained from Wuxi Tongwei feedstuffs Co., Ltd., Wuxi, China; ^2^ obtained from Hulunbeier Sanyuan Milk Co., Ltd., Inner Mongolia, China; ^3^ obtained from Yinhe Dextrin Co., Ltd., Zhengzhou, China; ^4^ obtained from Yifeng Food Additives Co., Ltd., Shanghai, China; ^5^ obtained from Jiangsu Fuyuda Food Products Co., Ltd., Yangzhou, China.

**Table 2 antioxidants-11-00228-t002:** Sequences of the primers used in the study.

Gene	GenBank Acc. No.	Primer Sequences (5′–3′)	Length (bp)	Purpose	Reference
T7-*relish*	KR827675.1	TAATACGACTCACTATAGGGATCATCATGAGGCGGAAAAG	40	RNAi	
		TAATACGACTCACTATAGGGTGGCATGTAGGTGAAATCCA	40		
T7-*dorsal*	KX219631.1	TAATACGACTCACTATAGGGCAAGTGTTCCTCGAAGGCTC	40	RNAi	
		TAATACGACTCACTATAGGGAACTTCACCAATTTGTCCGC	40		
*relish*	KR827675.1	GATGAGCCTTCAGTGCCAGA	20	RT-qPCR	
		CCAGGTGACGCCATGTATCA	20		
*dorsal*	KX219631.1	TCAGTAGCGACACCATGCAG	20	RT-qPCR	
		CGAGCCTTCGAGGAACACTT	20		
*imd*		CGACCACATTCTCCTCCTCCC	21	RT-qPCR	[24]
		TTCAGTGCATCCACGTCCCTC	21		
*toll*	KX610955.1	TTCGTGACTTGTCGGCTCTC	20	RT-qPCR	
		GCAGTTGTTGAAGGCATCGG	20		
*β-actin*	AY651918.2	TCCGTAAGGACCTGTATGCC	20	RT-qPCR	
		TCGGGAGGTGCGATGATTTT	20		

## Data Availability

Data is contained within the article.

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
