# Peer review of "High-Fat-Diet-Induced Oxidative Stress in Giant Freshwater Prawn (Macrobrachium rosenbergii) via NF-κB/NO Signal Pathway and the Amelioration of Vitamin E"

_antioxidants, 2022, doi:10.3390/antiox11020228_

Round 1

Reviewer 1 Report

The authors investigated effects of fat and vitamin E content in food on growth and stress parameters of crustacean species Macrobrachium rosenbergii. Since brine shims are an important part of aquatic ecological systems and are used in food industry, the topic of this manuscript is important and timely.

The authors explored biochemical parameters of haemolymph and hepatopancreas. The parameters were investigated by adequate and well-established methods. The results are supported by experimental data and they are well presented. Furthermore, the authors discussed the most important aspects of the results. Finally, the authors cited in the text also important literature.

Minor issue:

For readers, who are not familiar with physiology and anatomy of crustacean species, I would recommend a brief description of properties of crustacean haemolymph and also hepatopancreas in the introduction. I guess that about 5 sentences can be sufficient.

Reviewer 2 Report

this is a very great paper performed by the chineese , Ministry of Agriculture and Rural Affairs. 

this study found that long-term intake of high fat will accumulate in the body, leading to obstacles in fat transport, affecting body tissue fat, and reducing body fat deposition and fat metabolism by regulating lipid metabolism.

the question is: what are the translational effects in humans? the author did not discuss this issue.

In addition the authors, discussed that high fat diet caused excessive lipid deposition in M. rosenbergii, which further induced hepatopancreas lipophagy, apoptosis and lipid peroxidation, resulting in oxidative stress damage. 

I am still concerned: what is the additional importance of this paper?

Too many missing questions that authors and the editor must answer

Round 2

Reviewer 2 Report

the authors improved a lot the paper in line with my previous comments.

I found the paper acceptable for the publication in this current format

best

simone